# Downregulation of Zebrafish Cytosolic Sialidase Neu3.2 Affects Skeletal Muscle Development

**DOI:** 10.3390/ijms241713578

**Published:** 2023-09-01

**Authors:** Daniela Zizioli, Silvia Codenotti, Giuliana Benaglia, Marta Manzoni, Elena Massardi, Alessandro Fanzani, Giuseppe Borsani, Eugenio Monti

**Affiliations:** 1Unit of Biotechnology, Department of Molecular and Translational Medicine, University of Brescia, Viale Europa 11, 25123 Brescia, Italy; daniela.zizioli@unibs.it (D.Z.); silvia.codenotti@unibs.it (S.C.); giuliana.benaglia@unibs.it (G.B.); marta.manzoni@unibs.it (M.M.); alessandro.fanzani@unibs.it (A.F.); 2Unit of Biology and Genetics, Department of Molecular and Translational Medicine, University of Brescia, Viale Europa 11, 25123 Brescia, Italy; elena.massardi1@unibs.it (E.M.); giuseppe.borsani@unibs.it (G.B.)

**Keywords:** sialidases, zebrafish, muscle development, sialic acid, muscle differentiation, somite formation, myoblast differentiation

## Abstract

Sialidases remove terminal sialic acids residues from the non-reducing ends of glycoconjugates. They have been recognized as catabolic enzymes that work within different subcellular compartments and can ensure the proper turn-over of glycoconjugates. Four mammalian sialidases (NEU1-4) exist, with different subcellular localization, pH optimum and substrate specificity. In zebrafish, seven different sialidases, with high homology to mammalian counterparts, have been identified. Zebrafish Neu3.2 is similar to the human cytosolic sialidase NEU2, which is involved in skeletal muscle differentiation and exhibits a broad substrate specificity toward gangliosides and glycoproteins. In zebrafish *neu3.2*, mRNA is expressed during somite development, and its enzymatic activity has been detected in the skeletal muscle and heart of adult animals. In this paper, 1–4-cell-stage embryos injected with *neu3.2* splice-blocking morpholino showed severe embryonic defects, mainly in somites, heart and anterior–posterior axis formation. *Myog* and *myod1* expressions were altered in morphants, and impaired musculature formation was associated with a defective locomotor behavior. Finally, the co-injection of *Neu2* mouse mRNA in morphants rescued the phenotype. These data are consistent with the involvement of cytosolic sialidase in pathologies related to muscle formation and support the validity of the model to investigate the pathogenesis of the diseases.

## 1. Introduction

Sialidases (EC 3.2.1.18) are glycoside hydrolases that remove sialic acid residues from oligosaccharides, gangliosides and glycoproteins. Widely studied in viruses, microorganisms and vertebrates, several sialidases have been detected with peculiar subcellular localization and biochemical features. Since 1993 [1], the molecular cloning and expression of cDNAs encoding the different members of the sialidase gene family have allowed an impressive increase in the knowledge about these enzymes [2]. In mammals, four sialidases have been characterized, namely NEU1, 2, 3 and 4, and some of them are involved in various biological functions, often through their action on gangliosides and glycoproteins, with relevant implications in physiopathology [3].

For example, the expression of the cytosolic sialidase NEU2 in rat L6 myoblasts increases upon myotube formation induced by serum depletion, and the process is blocked by the addition of antisense oligonucleotide to the first eight codons of the cytosolic enzyme [4]. The same enzyme stably transfected in murine C2C12 myoblasts leads to spontaneous myoblast differentiation under standard growth conditions [5] and, finally, cytosolic sialidase transcript and enzyme activity increase in hypertrophic myofibers, whereas the effect is opposite upon the induction of myofiber atrophy [6]. Overall, these results demonstrate the involvement of NEU2 in the early phase of muscle differentiation, as well as in the phenomena underlying sarcomere remodeling, although the specific biochemical mechanisms by which the enzyme acts are still unknown.

The availability of zebrafish (*Danio rerio*) as a powerful animal model to study the in vivo biology of mammalian proteins prompted the identification and characterization of the sialidase gene family in this organism [7]. Seven different genes have been identified and, among them, *neu1* and *neu4* represent the orthologues of the mammalian sialidase *NEU1* and *NEU4*, respectively. The remaining five genes, namely *neu3.1*, *neu3.2*, *neu3.3*, *neu3.4* and *neu3.5*, form a cluster on chromosome 21, and all but *neu3.4* are transcribed during embryogenesis as demonstrated by RT-PCR. Intriguingly, whereas expression in the COS7 cells of Neu3.1, Neu3.3 and Neu4 shows that they are membrane-associated enzymes acting at an acidic pH on gangliosides, Neu3.2 is a soluble enzyme with a pH optimum of 5.6, which represent the ancestor of the mammalian cytosolic sialidase. A previous phylogenetic analysis of sialidase protein family in Metazoa revealed the presence of genes encoding cytosolic sialidase NEU2 in Amniota (mammals, birds and reptiles) [2]. The more recent and widespread application of next-generation sequencing (NGS) technologies for the study of the genome of living species led to the identification of putative NEU2 orthologs also in some species of amphibians (as reported in the NCBI Orthologs database), indicating that the gene likely originated in a common ancestor of Tetrapoda.

Recently, Neu1-knockout zebrafish has been obtained through the CRISPR/Cas9 genome-editing technique [8]. The morphological and molecular phenotypes of *neu1*-KO fish are rather similar to those detectable in Neu1-KO mice and in patients affected by sialidosis (OMIM # 256550). Overall, these results highlight again how the use of zebrafish to generate *neu1*-KO is of great utility for the study of this lysosomal sialidase in physio/pathological conditions. 

Starting from this multiple evidence, and trying to fill the gap between in vitro and in vivo experiments, we report here the results obtained in zebrafish by transiently switching off *neu3.2* gene expression using morpholino oligonucleotides and taking advantage by the important knowledge available on muscle formation and biology in this model organism. Moreover, an update was conducted on the presence of putative *neu3.2* orthologs in bony fish and its expression profile is conducted. Our in vivo results demonstrate that Neu3.2 is involved in the development of somites during zebrafish embryogenesis, with severe phenotypes in Neu3.2-deficient embryos regarding skeletal and heart formation that led to a defective locomotor behavior. 

## 2. Results

### 2.1. In Silico Analysis

The peculiar features of the zebrafish *neu3.2* gene prompted us to identify the presence of putative orthologs in other species of fishes. Considering the high level of sequence identity among *neu3*-like genes and the vast amount of genomic data currently available for vertebrate species, this was not straightforward task. For example, the NCBI lists genome data (including genome, transcript and protein sequence) for 1979 Actinopterygii (ray-finned fishes), the largest class of fishes. In most cases, only raw genome sequence data are available without any bioinformatic analysis presented in graphical interfaces, such as Genome Browsers.

To identify putative *neu3.2* orthologs, we analyzed the *neu3.2* entry in the NCBI Gene DB where a link leads to the NCBI Orthologs page that lists nine gene entries, all belonging to the *Cypriniformes* order of Teleostei (Appendix A). A multiple sequence alignment among the nine amino acid RefSeq sequences using the muscle algorithm revealed a sequence identity to zebrafish Neu3.2 ranging from 69% to 94%.

A BLASTN search performed with the nucleotide sequence of the *Danio rerio neu3.2* open reading frame identified several putative orthologs in the *Cypriniformes* order.

A BLASTP search performed after excluding the sequences in the NCBI non-redundant protein database that do not belong to the *Cypriniformes* failed to identify any bona fide Neu3.2-like polypeptide, indicating that the encoding gene emerged during evolution in this order of Teleostei.

RNA-level expression data for *neu3.2* and the all the other zebrafish sialidase genes were obtained from a systematic study performed using RNA-Seq reads from 18 different developmental stages of the zebrafish embryo [9]. The RNA-seq data indicate *neu3.2* is a maternally expressed gene that is also transcribed after zygotic genome activation becoming at most stages the most highly expressed sialidase gene after *neu1* (Appendix A).

### 2.2. neu3.2 Expression during Zebrafish Development and Soluble Sialidase Activity during Embryogenesis and in Adult Tissues

To analyze the spatio-temporal expression of maternal and zygotic *neu3.2* mRNAs during zebrafish development, whole-mount in situ hybridization (WISH) was performed on embryos from 0.2 to 48 h post-fertilization (hpf) using a digoxigenin-labeled antisense *neu3.2* mRNA probe of 700 nt. At the eight-cell stage (1 hpf), the transcripts were restricted in the blastodisc area and expressed in the blastodisc until reaching 90% epiboly. At 24 and 48 hpf, *neu3.2* was expressed in the developing nervous system and somites (Appendix A). To assess the expression levels of *neu3.2*, the soluble sialidase activity was measured in the soluble fraction obtained by whole-embryo extracts and using 4MU-NeuAc as a substrate at a pH of 5.6, as already described [7].

Figure 1 shows that, from 12 to 120 hpf, the soluble sialidase activity was always detectable in the time interval considered, with high levels in early stages, up to 16 hpf, and at 72 hpf. By considering the soluble sialidase activity detectable in the ovary, brain, heart and muscle apparatus of adult zebrafish, Neu3.2 was detectable in all the samples and with specific activities roughly superimposable with the values observed during embryogenesis. Interestingly, the zebrafish soluble sialidase activity detectable in the ovary and muscle was roughly double to that in the brain and heart.

Noteworthy, regarding *Neu2* transcript levels in high vertebrates, *Mus musculus* has medium expression levels in the muscle and skin and low values in the brain, lung, liver, colon, bone, brown and white adipose tissues, trachea, tongue, testis and kidney (https://www.ebi.ac.uk/gxa/genes/ensmusg00000079434?bs=%7B%22mus%20musculus%22%3A%5B%22ORGANISM_PART%22%5D%7D&ds=%7B%22kingdom%22%3A%5B%22animals%22%5D%7D#baseline), whereas in *Homo sapiens*, they are generally undetectable in all tissues but the skin (https://www.proteinatlas.org/search/NEU2). *Mus musculus* and *Homo sapiens* expression data have been accessed on 1 August 2023.

### 2.3. Silencing of neu3.2 by Morpholino Leads to Morphology Alterations in Muscles

In order to assess *neu3.2*’s functional role, we conducted knockdown experiments using a specific splice-blocking morpholino (*neu3.2*-MO). First, as a control for off-target effects, we performed a dose-response curve of *neu3.2*-MO and the standard morpholino (STD-MO) by injecting different doses into zebrafish embryos at the one–four cell stage (see Section 4). At 48 hpf, we evaluated the mortality and, whereas not significant differences were observed between non-injected (NI) and STD-MO-injected embryos at all the doses evaluated, starting from 1.5 pmol/embryo of neu3.2-MO injection, we observed a significant increase in mortality that exceeded 40% of embryos (Appendix A). Based on these results, we selected 1 pmol/embryo as the optimal dose for further experiments. The effect of *neu3.2*-MO in altering the correct splicing of the primary gene transcript is shown in Appendix A. 

We then carefully studied the morphology of the embryos injected with neu3.2-MO and those injected with STD-MO at 24 and 72 hpf (Figure 2). First of all, almost all of the STD-MO-injected embryos (>96%) had normal phenotypes, while the *neu3.2* morphants showed relevant morphology variations. We classified the observed phenotypes of morphants as follows: normal phenotype (NI and STD-MO); mild phenotype reduced head size and/or delay in development; severe phenotype body not developed, severe malformation in somites and presence of cardiac edema. As shown in Figure 2, more than 50% of Neu3.2-MO embryos showed severe phenotypes at 24 hpf and this value further increased at 72 hpf, with the remaining morphants being affected by a mild one. To avoid the non-specific activation of p53 [10], a p53-MO was co-injected with *neu3.2*-MO, and the phenotypes observed were similar to those obtained after the co-injections of *neu3.2*-MO alone (not shown). These data indicate that phenotypes induced by *neu3.2*-MO were not due to the non-specific activation of *p53* gene. 

These results strongly suggest a possible role of Neu3.2 in somite formation.

### 2.4. Rescue of the Phenotype

To prove the specificity of the morphant phenotype, we co-injected the *neu3.2*-MO together with *Mus musculus Neu2* mRNA. The dose-response curve of mouse *Neu2* mRNA is shown in Appendix A. By co-injecting 100 pg/embryo *of Mus musculus Neu2* mRNA in *neu3.2*-MO morphant embryos, we observed that 72% of morphants recovered the normal phenotype, thus underlining the conservation of vertebrate soluble sialidases function during evolution. As expected, *neu3.2* morphants show a strong reduction (80%) in Neu3.2 enzymatic activity (Figure 3). 

### 2.5. Knockdown of neu3.2 Impairs Cardiac Development in Zebrafish Embryos 

To investigate the role of *neu3.2* during heart development, we evaluated the cardiac structure of *neu3.2* morphants by whole-mount in situ hybridization (WISH), with the *cmlc2* heart-chamber marker known to be mainly expressed in the ventricle and at lower level in the external curvature of the atrium (Figure 4). 

At 36 hpf, when the heart begins to loop in the control embryos, the heart tube appeared relatively straight and smaller in *neu3.2* morphants. The analysis of the looping phenotypes by the visualization of the developing atrium and ventricle was performed at 48 hpf using the transgenic line tg:*Bmp*-EGFP [11]. Bone morphogenic protein (BMP) signaling has important roles during vertebrate cardiovascular development [12], and thus its expression in the zebrafish transgenic line represents a useful tool to study heart embryonic development. In the control embryos (STD-MO), the heart structure appeared phenotypically indistinct with the ventricle and atrium overlapping each other. In contrast, *neu3.2* morphants showed an elongated, string-like heart tube, displaying a reduced and morphologically-altered ventricle. A functional analysis was performed by counting the heart beats per minute; we observed a reduction of 30% in heart beats in *neu3.2* morphants compared to the control embryos. These data suggest a possible developmental delay, but also a potential heart-looping failure, supporting the role of *neu3.2* in cardiac muscle development. 

### 2.6. Neu3.2 Involvement in Muscle Development

It has been reported that either *myf5* or *myod1* is sufficient to promote slow muscle formation from adaxial cells and that *myod1* is required for fast muscle differentiation [13]. Muscle development, which starts with the generation of two fiber types in the embryo followed by the addition of three more types in mature fish, is well characterized at the cellular level, and the underlying molecular signals are beginning to be elucidated [14]. To assess the requirement for *neu3.2* during embryonic myogenesis, we performed a WISH analysis with the somite-specific marker *myod1*. At the bud stage (14 hpf), the expression of *myod1* in the adaxial cells was reduced in *neu3.2* morphants, particularly at the posterior end (Figure 5). 

This was more evident in *neu3.2* morphants at 22 hpf, where it is possible to observe the downregulation of *myod1* expression in somites and severe abnormalities of less distinct somites compared to the non-injected embryos, suggesting impaired somite organization together with shortened anterior/posterior axes and undulated notochord. More than 85% of *neu3.2* morphants with the severe phenotype and 60% with the mild phenotype had to be manually dechorionated (Figure 5). This delay in hatching could be due to an overall developmental delay but also to a reduction n the muscle activity that contributes to the natural exit of the embryos from their protective outer chorion. Confocal analysis shown in Figure 5, panel C, confirmed the results previously obtained in *cmcl2* WISH both in *neu3.2* morphants showing mild phenotype and in those with a severe phenotype, indicating that the fibers are not well formed and organized. 

### 2.7. Neu3.2 Deficiency Leads to Abnormal Myogenesis 

To fully appreciate at a microscopic level the phenotype of *neu3.2* morphants, we performed a staining with phalloidin conjugated with a fluorescent dye of embryos at 48 hpf. In *neu3.2* morphants, the myosectum was difficult to distinguish, and in some regions, it appeared interrupted, while in STD-MO-injected embryos, the fibers were well defined and structurally intact (Figure 6). 

The myofibers appeared misaligned with markedly disorganized shape and orientation, with respect to the controls. The NI embryos showed well-organized myofibers. To evaluate the effect of *neu3.2* knockdown during myogenesis, we analyzed the expression profile by real-time PCR of three genes with important roles during early myogenesis: *myod1*, *myf5* and *myog*. At the early stage of development (12–14 hpf), when the muscle formation begins, all three markers are downregulated in *neu3.2* morphants compared to the controls (Figure 7).

### 2.8. Downregulation of Neu3.2 Function Causes Locomotor Hypoactivity 

Given the severe impairment of muscle development and organization, we completed the phenotypic analysis of *neu3.2* morphants by investigating their motor behavior at different stages, which can be easily evaluated in this model system. Embryos at 48 hpf exhibit rapid acceleration and deceleration in movement/swimming in response to visual, tactile or auditory stimuli. The startle response is important because it provides an integrated state of sensory and motor stimuli and can be induced by touching the head or tail of zebrafish larvae [15]. We first investigated the consequences of decreasing Neu3.2 activity on embryos at 48 hpf: only a small percentage of morphants (32%) showed a response in the “touch-evoked test” compared to the non-injected embryos (90%) as well as embryos injected with *Mus musculus Neu2* mRNA (70%), confirming the specificity of the phenotype. At 72 hpf, we observed a strong decrease in tail coil movements in *neu3.2* morphants compared to the embryos that were not injected or injected with *Mus musculus Neu2* mRNA. Changes in the frequency of tail coil activity may depend on defective early myogenesis and neurogenesis [16]. Zebrafish larvae start swimming at 4–5 dpf once they have a swim bladder, and the movement is a behavior due to an interplay between developing neurons and somites. Single 6-day-old morphants and wild-type larvae were placed in the wells of a 96-well plate and the total distance travelled was registered for 120 min according to a well-established protocol [17]. We observed a significant decreased in the distance swum by *neu3.2* morphants compared to that of the non-injected embryos. Similar results in locomotion were observed in embryos injected with *Mus musculus Neu2* mRNA (Figure 8) [18].

## 3. Discussion

Sialidases in bony fishes have been identified in several species, such as *Danio rerio* [7], *Oryzias latipes* [19] and *Oreochromis niloticus* [20], all living in fresh water and originally found in the Indian subcontinent, Japan and various lakes and rivers in Africa, respectively. In *Homo sapiens*, the gene encoding the lysosomal sialidase NEU1 is responsible for sialidosis (https://omim.org/entry/256550). Fish animal models have allowed the study of this genetic disease, with knockout animals showing phenotypes similar to Neu1-KO mice and encompassing the clinical features typical of the patients [8]. Moreover, the molecular characterization of fish sialidases have allowed the discovery of the roles played by the other gene family members in important biological phenomena both in vitro and in vivo. For example, the transfection of *neu3a* from *Oryzias latipes* in mouse neuroblastoma cells induces the positive regulation of retinoic-acid-induced differentiation, with an increase in the axon length [19]. An effect on neurogenesis has been described also for *Oreochromis niloticus* Neu4 [21] with the enzyme involved in retinal development in vivo, and its transfection in mouse and human neuroblast cell lines increased neurite formation. 

Based on the results obtained in vitro upon the involvement of the cytosolic sialidase in myoblast differentiation into myotubes and summarized in the Introduction, we used zebrafish as an animal model to evaluate the role of this peculiar enzyme in vivo. Our results demonstrate that, upon *neu3.2* downregulation, muscle formation is impaired during the early phase of embryogenesis, and relevant consequences on motility behavior are detectable in the developing fish. In addition, the downregulation of this soluble sialidase has relevant consequences on cardiac development and function. Briefly, *neu3.2* morphants showed an elongated, subtle heart tube with a small and altered ventricle, and the functional analysis revealed a significant reduction in the heart rate of morphants compared to the control embryos. Moreover, the phenotypes observed directly depend on cytosolic sialidase as demonstrated by the phenotype rescue, achieved upon the co-injection of *Mus musculus Neu2* mRNA in the *neu3.2* morphant embryos. 

Our in vivo results are in agreement with those obtained by studying Neu3b from *Oryzias latipes* that show that Neu3.2 from *Danio rerio* has a cytosolic localization and thus behaves as the mammalian NEU2 [22]. In addition, the *neu3b* gene is highly expressed at the post-hatching stage, a period during embryogenesis that is very important for muscle formation [23], and the overexpression of the fish soluble sialidase in mouse C2C12 myoblasts accelerates differentiation into myotubes upon serum deprivation. As observed in our experiments in vivo, the in vitro formation of myotubes also occurs with the up-regulation of *myod1* and *myog* transcripts, two important myogenesis biomarkers [24]. Compared to the control cells, Neu3b overexpression modifies the ganglioside composition of C2C12 myoblasts, mainly with a decrease in ganglioside GM2 and an increase in Lac-Cer. No significant variation is detectable on the glycoprotein pattern as demonstrated by Western blot and lectin specific for sialic acid. Intriguingly, a significant decrease in the *egfr* mRNA level is detectable in *neu3b*-transfected C2C12 cells, and EGFR downregulation is one of the events responsible for positive myogenesis regulation, especially at the early stage of myogenesis. Another possible target of Neu3b as an enzyme that stimulates myogenesis is represented by the neural cell adhesion molecule (NCAM), a glycoprotein conjugated with polysialic acid (PSA) involved in muscle formation [25]. PSA contains sialic acid residues linked with a (2–8) linkage and Neu3b recognizes colominic acid as substrate, which is a polymer of SA with this particular glycosidic linkage [19]. Thus, the desialylation of PSA linked on NCAM by Neu3b could represent another crucial point because myotube formation is inhibited by high PSA levels. To correlate this evidence obtained in vitro with *Oryzias latipes* Neu3b and our in vivo results obtained with the downregulation of *Danio rerio* Neu3.2, we studied the substrate specificities of the latter enzyme. This study was conducted on purified, recombinant Neu3.2 produced in *E. coli* and reveals interesting information about the biochemical features of the soluble enzyme [26]. All the gangliosides tested are substrates, and Neu3.2 recognizes better ganglioside GM1, followed by GD3, GD1a and GM3. The soluble sialidase removes sialic acid residues from the glycoproteins fetuin and mucin as well as from oligosaccharide sialyllactose with high efficiency. Overall, these results indicate that Neu3.2 has a broad substrate specificity at least used in in vitro assays, and thus most of the mechanisms involved in C2C12 myoblast differentiation upon Neu3b transfection should have an implication in our in vivo results upon *Danio rerio* cytosolic sialidase downregulation.

Concerning C2C12 myoblast differentiation, a study on glycogenome expression pointed out how the myoblast cell membrane and ECM could be modified for cell fusion, leading to myotubes [27]. Among the genes upregulated, the relative quantity of *St8sia5* mRNA increases more than 1300-fold as detected at 72 h of differentiation. The gene encodes ST8 alpha-N-acetyl-neuraminide alpha-2,8-sialyltransferase 5, an anabolic enzyme involved in the synthesis of gangliosides GD1c, GT1a, GQ1b and GT3 (https://www.proteinatlas.org/ENSG00000101638-ST8SIA5). In the same cluster of upregulated genes, although with a much lower mRNA increase, corresponding to roughly 19-fold, *Neu2* also encodes cytosolic sialidase, which with its catabolic action contributes to the regulation of cell sialic acid content. Overall, these findings further underline the relevance of sialoconjugates in complex phenomena, such as cell differentiation, adhesion, recognition and fusion. 

Recently, another study revealed interesting results on cytosolic sialidase Neu2 biology [28]. The study was performed on a *Neu2* knockout mouse model generated using the CRISPR/Cas9 genome-editing technology, and the offspring littermates were analyzed at different ages, namely young and adult, looking for in vivo function(s) of the soluble members of the sialidase enzyme family. Notably, the lack of NEU2 abolished the lipid metabolism with severe, generalized consequences in homozygous knockout mice, such as serum hyperlipidemia, the lipid accumulation in the liver that causes severe steatosis in elderly animals. In addition, the abrogation of the lipid metabolism impaired muscle differentiation and morphology. Actually, it is well known that skeletal muscle utilizes energy that also derives from the lipid metabolism, both at rest and during contraction. In *Neu2* homozygous KO mice, a significant decrease in myofibers and perimysium was detectable in the extensor digitorum longus (EDL) muscle, with an increase in the nuclear Myod1-positive cells leading to impaired muscle fiber differentiation and maturation. As expected, plasma hyperlipidemia leads to the accumulation of long-chain fatty acids in myotubes and a decrease in glucose uptake [29], with important additional consequences on the supply of oxidable fuels for energy metabolism. Thus, *Neu2* KO mice showed impairments in muscle performance with a deterioration that reached the maximum values in the elderly group. The measurements of the energy expenditure rate demonstrated a reduction in *Neu2* KO mice compared to the control wild-type littermates and results in an obese phenotype with significant increases in the body weight. Finally, a detailed glycoproteomic analysis revealed a number of glycoproteins with altered sialic acid content in *Neu2* KO mice that are associated to the lipid metabolism and muscle function, identifying several targets for further studies in this research field. Overall, despite the evolutionary distance between the animal model used, namely teleost, and mammals, our results in *Danio rerio* match those described in *Mus musculus* [28] and pave the road for the more detailed biochemical and functional characterization of zebrafish *neu3.2* morphants. 

Finally, the results described in this paper further confirm the pivotal role played by the cytosolic sialidase in muscle differentiation and development, providing further information about sialidase biology and the relevance of sialic acid in regulating a great number of pivotal physiological processes.

Further experiments are in progress in our laboratory in order to better understand the biochemical basis of the intriguing muscle phenotype observed in *Danio rerio neu3.2* morphants.

## 4. Materials and Methods

### 4.1. Bioinformatic Analysis

Zebrafish genomic sequences were analyzed using the University of California Santa Cruz (UCSC) Genome Browser (http://genome.ucsc.edu/) on the GRCz11/danRer11 (May 2017) *Danio rerio* genome assembly. Nucleotide and amino acid sequences were compared to the non-redundant sequences present at the NCBI (National Center for Biotechnology Information) using the Basic Local Alignment Search Tool (BLAST) [30]. Multiple sequence alignment was performed using the MUSCLE algorithm [31].

### 4.2. Zebrafish Maintenance 

Zebrafish were maintained and used following protocols approved by the Local Committee for Animal Health (OPBA, *Organismo Per il Benessere Animale*, No. 211B5-10) and in accordance with the Italian and European regulations on animal use. The following strains were used: the wild-type zebrafish AB strain and the transgenic lines Tg neurod:EGFP [32] and Tg (Bmp:EGFP) [11].

### 4.3. Fish Breeding and Embryo Collection 

Adult zebrafish were all kept in tanks containing 3–5 L of water at 28 °C on 14 h light/10 h dark cycle [33]; they were bred by natural crosses and the collected embryos were staged according to Kimmel et al. [34]. Immediately after spawning, the fertilized eggs were incubated at 28 °C in 10 cm Ø Petri dishes in fish water (0.1 g/L Instant Ocean Sea Salts, 0.1 g/L sodium bicarbonate, 0.19 g/L calcium sulfate and 0.2 mg/L methylene blue in H_2_O) until the desired developmental stage was reached. For the whole-mount in situ hybridization (WISH) experiments to examine post-gastrulation stages, beginning from 24 hpf, regular fish water was replaced by a 0.0045% 1-phenil-2-thiourea-PTU (Sigma-Aldrich, Saint Louis, MO, USA) solution. The embryos were dechorionated by hand using sharpened forceps and then fixed in 4% (*w*/*v*) paraformaldehyde 1× PBS overnight at 4 °C (or 2 h at room temperature) in 2 mL tubes; they were dehydrated through sequential washes in 25%, 50%, 75% methanol/PBS and 100% methanol and stored at least over night at 20 °C. For the extraction of total RNA or enzymatic assay, the embryos were deprived of all water and stored at −80 °C until processing.

### 4.4. Morpholino Microinjection

To knockdown the expression of zebrafish Neu3.2 protein, we designed a specific splicing-inhibiting morpholino (Gene Tools, LCC) (5′-ATTATTGTTGAAAGCCTCACCTCGA-3′) that targeted to the exon 2/intron 2 splice boundary of transcript neu3.2-001 (ENSDART00000003518) (membrane sialidase), tandem duplicate 2, gene (neu3.2). The designed morpholino caused the activation of a cryptic site splicing, thus altering the translation reading frame of exon 2 with the introduction of a premature stop codon (Appendix A). As the control, we used a standard (non-specific, ST-MO) morpholino oligonucleotide (5′-TTTACAAGACCGTCTACCTTTCCAA-3′) (Gene Tools, LCC). Different amounts of morpholino (0.3, 0.5, 0.7, 1, 1.5 and 2 pmol/embryo) were injected into wild-type embryos at 1–2 cell stages to determine the optimal concentration for the knockdown experiments. The injected embryos were analyzed at 48 hpf for mortality and morphology, and a dose-response curve was established (Appendix A). The final concentration of 1 pmol/embryo (corresponding to 8 ng/embryo) was used for all experimental procedures. The morpholinos were introduced in 1X fish water together with dye-tracer red phenol. A standard control (non-specific, ST-MO) morpholino oligonucleotide (5′-TTTACAAGACCGTCTACCTTTCCAA-3′) was used as the negative control (Gene Tools, LCC). After the microinjection, the embryos were incubated in 1X fish water supplemented with 0.003% PTU at 28 °C to prevent pigmentation processes. As a control, we used embryos injected with the standard morpholino (ST-MO). For the injection experiments, we used an Eppendorf FemtoJet Micromanipulator 5171 (Eppendorf AG, Hamburg, Germany). The injected embryos were then collected in Petri dishes and maintained in fish water at 28 °C. To confirm the targeting efficacy of the *neu3.2*-MO, an alternative splicing pattern analysis was performed on the embryos’ cDNA, at 72 hpf, using the following primers: *neu3.2*spl F (5′-AGCTTCGTCTGTTCGTCTGA-3′) and *neu3.2*ZBfR3 (5′-TAAGAATCAGGCACGCCTTT-3′). 

### 4.5. Mus Musculus mRNA Synthesis and Injections for Phenotypic Rescue Experiments 

For rescue experiments on *neu3.2* morphants, we selected the oligonucleotide primers Mm Neu2 F (5′-atgaattcATGGCGACCTGCCCTGTCCTG-3′) and Mm Neu2 R (5′-atctcgagTCACTGGGCATCAAATACAGT-3′) to amplify the ORF of Neu2 from *Mus musculus* by RT-PCR. A total of 2 µg of total RNA were retrotranscribed with 400 units of MMLV-RT (Promega, Madison, WI, USA) for 1 h at 37 °C, and the cDNA was used for PCR amplification by Taq DNA polymerase (Promega). The PCR product was digested with Eco RI and Xho I and cloned in the pCS2+ vector digested with the same restriction enzymes. The automated sequencing of recombinant pCS2 + Neu2 constructs confirmed the sequence of the cloned inserts. The plasmid construct was linearized, the cloned insert was transcribed using QuantiTect Reverse Transcription Kit (QIAGEN, Hilden, Germany) according to the manufacturer’s instructions, and *Neu2* mRNA was diluted in distilled water to a final concentration of 1 µg/µL. The optimal dose for injection in the rescue experiments was determined by dose–response curve experiments performed in wild-type embryos. A total of 100 pg/embryo was the highest amount of injected mRNA endoding *Mus musculus Neu2* that did not induce phenotypic alterations. The rescue experiments were thus conducted by co-injecting *neu3.2*-MO together with 100 pg/embryo of *Mus Musculus Neu2* mRNA.

### 4.6. Real-Time PCR

Total RNA was isolated from 30 zebrafish embryos at different developmental stages (2.8 cells, high stage, 50% epiboly) using the TRIzol protocol (Thermo Fisher Scientific, Waltham, MA, USA) according to the manufacturer’s protocol, quantified using the mySPEC micro-volume spectrophotometer (VWR International, Philadelphia, PA, USA). and controlled by electrophoretic separation on 1.5% TAE agarose gel and ethidium-bromide-stained. cDNA was prepared from 1 µg of total RNA, using an Access RT-PCR System kit (Promega) in the presence of oligo-(dT) primers, following the manufacturer’s instructions. 

Real-time PCR was performed using the Eco-Illumina system. The reactions were performed in a 10 μL volume, with 0.5 μM of each primer (sequences in Table 1), 5 μL of SYBR Green Master Mix (Bio-Rad Laboratories S.r.l., Segrate, Italy) and 20 ng of cDNA. The amplification profile consisted of a denaturation program (95 °C for 1 min) and 40 cycles of two-step amplification (95 °C for 15 s and 60 °C for 30 s), followed by a melting cycle. Each reaction was performed in triplicate and *Dre rpl13a* was used as a reference gene. The relative levels of expression were calculated by the ΔΔCT method. 

PCR was conducted using the forward (F) and reverse (R) primers listed in Table 1.

### 4.7. neu 3.2 Riboprobe Synthesis

To synthesize the riboprobes for the detection of zebrafish *neu3.2* transcripts, we amplified specific regions by PCR using as template the cDNA from the ovary of the adult fish (9 months of age) and the oligonucleotide primers *neu3.2 spl*(5′-AGCTTCGTCTGTTCGTCTGA-3′) and BC077080ZR2 (5′-CTGTTTCGAGCATTGCAGTAAAGTT-3′). The amplification conditions were the following: 2 min at 94 °C, 40 cycles at 94 °C for 30 s, 53 °C for 30 s and 72 °C for 1 m 30 s, followed by a final extension at 72 °C for 5 min. The PCR products were subcloned in the pGEM-T-Easy System (Promega) and verified for the sequence and orientation of the inserts. In particular, *neu3.2* antisense and sense riboprobes were obtained by in vitro transcription of the cloned cDNAs with T7 or SP6 RNA polymerase, using a Digoxigenin Labeling mixture, according to the manufacturer’s instructions (Roche S.p.A., Monza, Italy). The probe length was 900 bp.

### 4.8. Whole-Mount In Situ Hybridization (WISH)

Whole-mount in situ hybridization was performed according to a standard method [35]. Zebrafish embryos were collected, dechorionated, fixed in 4% (*w*/*v*) paraformaldehyde/PBS overnight at 4 °C, rinsed twice in PBS/1% Tween 20 and then dehydrated in methanol and stored at −20 °C. Before WISH, the stored embryos were rehydrated through descending methanol series and, after treatment with proteinase K (10 μg/mL, Roche), the embryos were hybridized overnight at 68 °C with DIG-labeled antisense or sense RNA probes (500 ng/500 mL). The embryos were washed with an ascending scale of HybeWash/PBS and SSC/PBS, and then incubated with anti-DIG antibody conjugated with alkaline phosphatase overnight at 4 °C. The staining was performed with NBT/BCIP (blue staining solution, Roche) alkaline phosphatase substrates. When different types of probes (sense vs. antisense) or embryos (injected vs. non-injected) had to be compared, all incubations were conducted at the same time, at the same probe concentration and, when possible, with the same reagents and solutions. WISH images were taken with Zeiss Axio Zoom V16 equipped with Zeiss Axiocam 506 color digital camera and processed using Zen 3.5 (Blue Version) software from Zeiss (Carl Zeiss S.p.A., Milan, Italy). The developmental stages of the zebrafish embryos were expressed as hpf or dpf (days post-fertilization) at 28.5 °C [34]. 

### 4.9. Sialidase Extraction and Enzymatic Assays

Embryos at different stages and fresh adult organs were washed in PBS and harvested in 350 mL of 0.25 M sucrose/1 mM EDTA plus a mix of protease inhibitors (Roche); then, they were sonicated at 4 °C and centrifugated at 600× *g* for 10 min at 4 °C to remove the coarser parts. The supernatant was ultracentrifuged at 100,000× *g* for 1 h at 4 °C to obtain the soluble/cytosolic fraction that was used as an enzymatic source. As previously described, the soluble/cytosolic sialidase Neu3.2 had a pH optimum of 5.6 and the enzyme activity was measured using 4-MU-NeuAc (2-(4-methylumbelliferyl)-a-D-N-acetylneuraminic acid) (Sigma) as an artificial, fluorescent substrate. Briefly, the assay was performed with up to 10 µg of total proteins, and the reactions were set up in triplicate in a final volume of 200 µL with 0.2 mM 4-MU-NeuAc and 600 µg BSA with 12.5 mM sodium citrate/phosphate buffer and a pH of 5.6. After incubation at 37 °C for 30 min, the reaction was stopped by the addition of 0.8 mL of 0.25 M glycine/NaOH at a pH of 10.4. The amount of fluorescent 4-methylumbelliferone (4MU) released was determined fluorometrically using a Jasco FP-770 fluorimeter whit an excitation and emission wavelength of 365 nm and 445 nm, respectively, and 4-MU to set up a calibration curve [7]. 

### 4.10. Imaging

The bright-field imaging of the embryos and larvae (anaesthetized with tricaine 0.16 mg/mL embedded in 0.8% low-melting agarose and mounted on a depression slide) and WISH pictures were captured using a Zeiss Axio Zoom V16 equipped with Zeiss Axiocam 506 color digital camera and processed using Zen 3.5 (Blue Version) software from Zeiss (Carl Zeiss S.p.A., Milan, Italy). Confocal Images were acquired using a Plan-Neofluar 10X/0.3NA objective. For light sheet microscopy analysis, the embryos were first anesthetized using tricaine (0.02% in fish water) and subsequently included using a low-melting agarose matrix (Top Vision Low-Melting-Point Agarose, Thermo Fisher Scientific, Monza, Italy) at 0.5% in fish water. Images were acquired using Zeiss Light Sheet microscope V1 supported by ZenPro software using a 488 nm laser and 505–545 nm filter. Images from the same experiment were taken with the same laser intensity and exposure time to generate comparable images. After the acquisition, 3D images were generated and manipulated using Arivis Vision 4D (Zeiss, Oberkochen, Germany). Muscle imaging in embryos at 48 hpf was obtained using phalloidin [36] conjugated with iFluor 594 dye (ab176757, Abcam. Prodotti Gianni, Milan, Italy) and following the manufacturer’s protocol with the appropriate modifications to work with fish embryos.

### 4.11. Statistical Analysis

All the experiments described in the manuscript were performed at least two or three times using GraphPad Prism V8 (Dotmatics, Boston, MA, USA) to perform the statistical analyses. The comparison and significance between different groups were determined by a one-way ANOVA, corrected for multiple comparisons by two-tailed unpaired Student’s *t*-test. The *p*-value is indicated with asterisks: * *p* < 0.05, ** *p* < 0.01, *** *p* < 0.001 and **** *p* < 0.0001. Differences were considered significant at *p*-values of less than 0.05.

### 4.12. Behavioral Tests in Embryos: Touch-Evoked Tests and Tail Coil Spontaneous Movements

Zebrafish embryos were injected with *neu3.2*-morpholino and co-injected with *Mus musculus Neu.2* mRNA and embryos injected with the standard morpholino (STD-MO) were used as the negative control. The embryos were kept at 28 °C until they reached the correct stage of development to conduct the test. 

At 48 hpf, 40 embryos for each type were subjected to the touch-evoked test, following the protocol [37] with slight modifications. A motility wheel consisting of four concentric circles of increasing diameter (5, 10, 15 and 20 mm) was placed under the microscope and centered at the bottom of a 60 mm Ø Petri dish containing fish water. 

Each embryo was placed in the center of the inner circle and the tail was gently stimulated with a poker tool; the distance it swam in the predetermined concentric circles was recorded. If the embryo could not cross the first circle after multiple attempts (5 attempts), it was determined as incapable of exiting the circle. Once the data of the distance swam by all 25 embryos of each group were obtained, the percentage of embryos crossing each predetermined concentric circle were calculated. A total of 40 injected embryos (*neu3.2*-morpholino and co-injected with *Mus musculus Neu.2* mRNA and embryos injected with the standard morpholino (STD-MO) used as the negative control) were observed for head–tail coil spontaneous contractions for 1 min under microscope. The total number of head–tail coil movements (tail flip) of three independent experiments was recorded for each embryo, the average was calculated and a box diagram was plotted for the injected and non-injected embryos. 

### 4.13. Analysis of the Locomotor Behavior in neu3.2 Morphant Larvae

The experiments were performed essentially as previously described [38]. Locomotor activity was examined at 6 days post-fertilization (dpf): 12 surviving larvae for each experimental condition (injected with STD-MO, *neu3.2*-MO and co-injected *neu3.2*MO with Mm *Neu2* mRNA) were collected in a 96-square-well plate with one larva per well in a volume of 200 µL fish water 1X. The 96-square-well plate was then placed in the observation chamber of the Danio Vision Noldus system holder (Noldus, Wageningen, The Netherlands), in an isolated noise-free room. The larvae were allowed to adapt for 30 min before video recording. The system was then set up to track movements (moved distance in 2 min time bins) for 2 h by 6 cycles of alternating light and dark 10 min periods. Data were analyzed using the Noldus Ethovision software (Noldus, https://www.noldus.com/, accessed on 21 June 2023). Movements were reported as the total distance (cm) travelled by the larvae calculated under both light and dark stimuli. The data obtained were exported to Excel (Microsoft) for analysis. To compare the distance swum between the wild-type and morphant larvae, *p*-values were calculated using a one-way ANOVA followed by paired *t*-tests with unpooled SD and the *p*-value was adjusted according to HOLM. The experiments were repeated three times. 

## Figures and Tables

**Figure 1 ijms-24-13578-f001:**
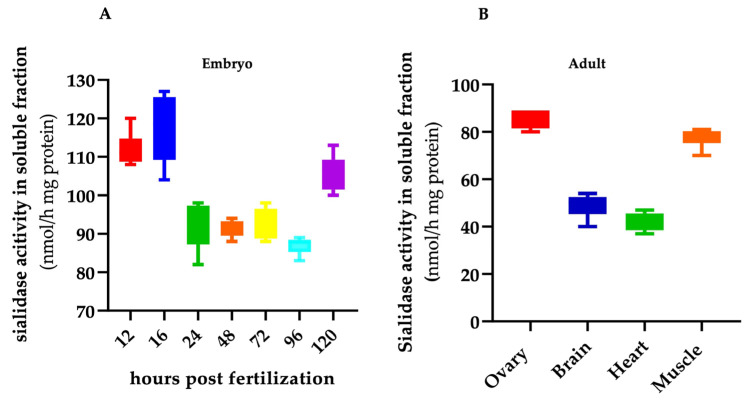
Soluble sialidase activity in zebrafish embryo and adult organs. Total extracts from wild-type embryos (**A**) and organs of 9-month-old adult fish (**B**) were assayed for soluble sialidase activity, as described in the Section 4 (*n* = 30). Extracts from various embryonic stages (12, 16, 24, 48, 72, 96 and 120 hpf) and from ovary, brain, heart and muscle were used as the enzyme sources. Assays were performed with up to 10 µg of total proteins. Sialidase activity was measured using 4-MU-NeuAc as a substrate at a pH of 5.6. Values are the mean ± SD of three independent experiments.

**Figure 2 ijms-24-13578-f002:**
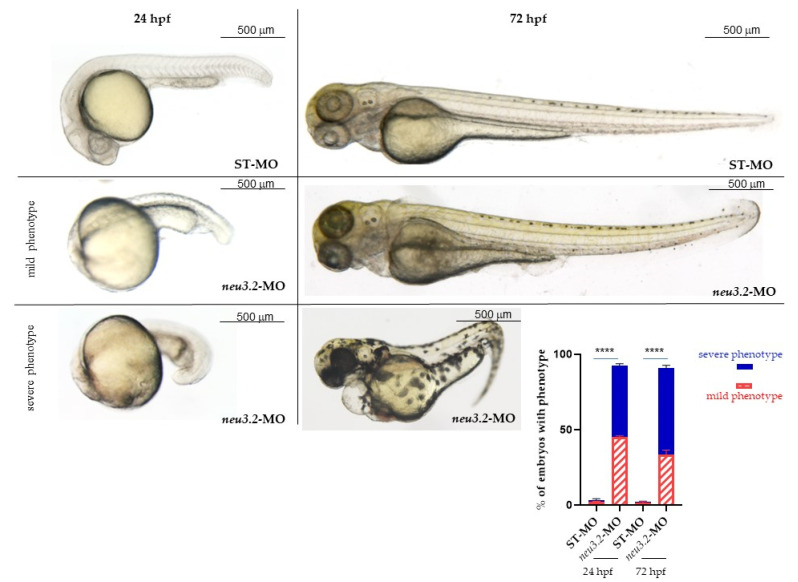
Effects of neu3.2 morpholino injection on zebrafish development. Representative images of the phenotype at 24 and 72 hpf obtained after the injection of neu3.2-MO. The injected embryos were compared with embryos injected with the standard morpholino (STD-MO). The embryos were injected with 1 pmol/embryo of *neu3.2*-splicing morpholino at 1–2 cell stages. For each injection experiment (*n* = 5), about 150 embryos were analyzed for morphology and classified with mild and severe phenotypes. The graphic in the bottom right indicates the percentage of embryos with mild and severe phenotypes. The results are expressed as mean ± SD of 5 independent experiments. (**** *p* < 0.0001 vs. injected embryos with standard morpholino-STD-MO—Unpaired; two-tailed *t*-test).

**Figure 3 ijms-24-13578-f003:**
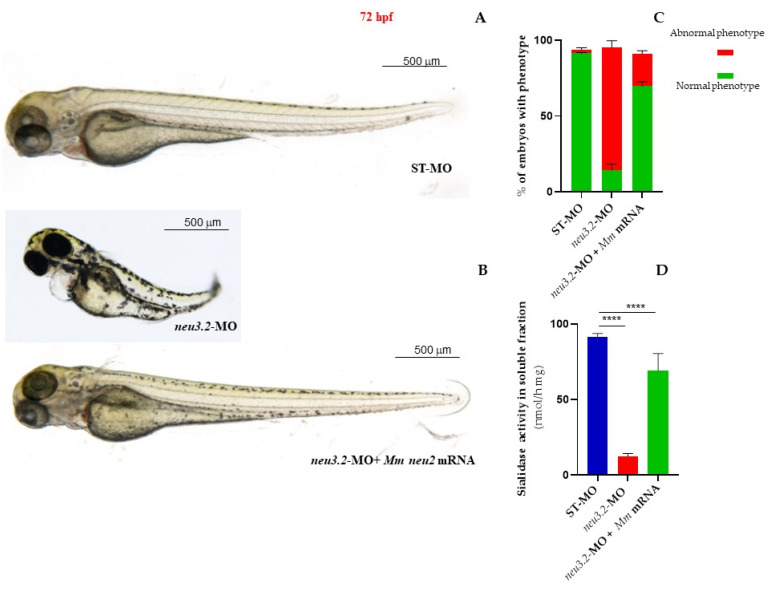
Rescue of the phenotype in *neu3.2* morphants with *Mus musculus* Neu2 mRNA. Representative images, at 72 hpf, of the injected embryos with STD-MO (**A**) and with neu3.2 MO (1 pmol/embryo) alone (**B**) or together with 50 pg/embryo *Neu2* mouse mRNA (**C**). The percentage of embryos with the different phenotypes (normal and severe) in the three categories of the analyzed embryos is shown in the graph (**C**). Neu 3.2 activity measured in the soluble fraction using 4-MU-NeuAc as a substrate at a pH = 5.6 in all three categories of analyzed embryos is shown in the graph (**D**). Three experiments were performed and more than 150 embryos of each type were analyzed. (**** *p* < 0.0001 vs. injected embryos with standard morpholino ST-MO—Unpaired; two-tailed *t*-test).

**Figure 4 ijms-24-13578-f004:**
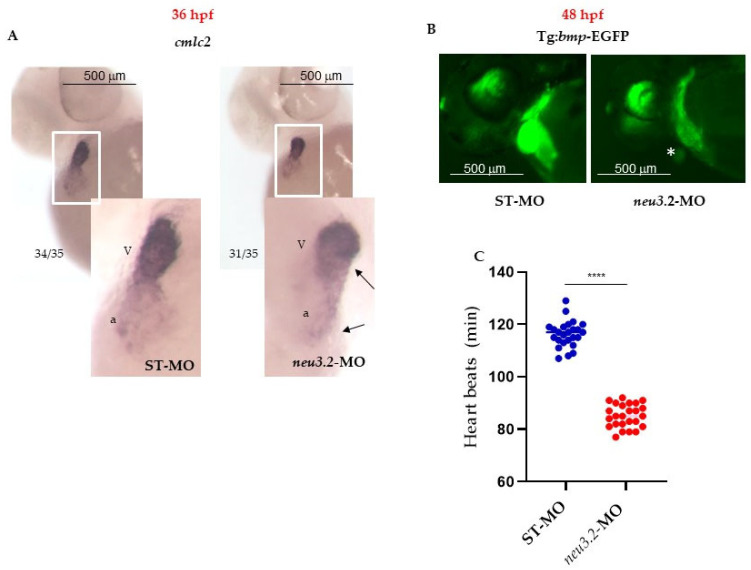
Downregulation of *neu3.2* affects cardiac development. (**A**) Analysis of the cardiac marker *cmlc2* by whole-mount in situ hybridization (WISH) in zebrafish embryos injected with STD-MO and injected with *neu3.2*-MO (1 pmol/embryo). Images are the anterior views of the whole mount with heart with dorsal up. Arrows point to the atrium (a) and ventricle (V), demonstrating their small size compared with the STD-MO-injected embryos. Ratios at the bottom left part of each picture specify the number of embryos showing the same staining pattern, compared to the total number of embryos used for each experiment. Two replicates were performed (*n* = 35). (**B**) Representative images of the cardiac area in the Tg(*Bmp*:EGFP) line STD-MO-injected and *neu3.2*-MO-injected (1 pmol/embryo at 48 hpf) embryos. Asterisk points out the reduction of fluorescence intensity and size of the heart in both conditions. Results are from one representative experiment with at least 25 embryos out of three independent replicates. (**C**) The graph shows the comparison of the heart rate (numbers of beats per minute) between STD-MO-injected and *neu3.2*-MO-injected (1 pmol/embryo) embryos and it is representative of one experiment (*n* = 25). The experiment was repeated three times. Results are expressed as the mean ± SD of three independent experiments. (**** *p* < 0.001 vs. the control group—embryos injected with standard morpholino-STD-MO).

**Figure 5 ijms-24-13578-f005:**
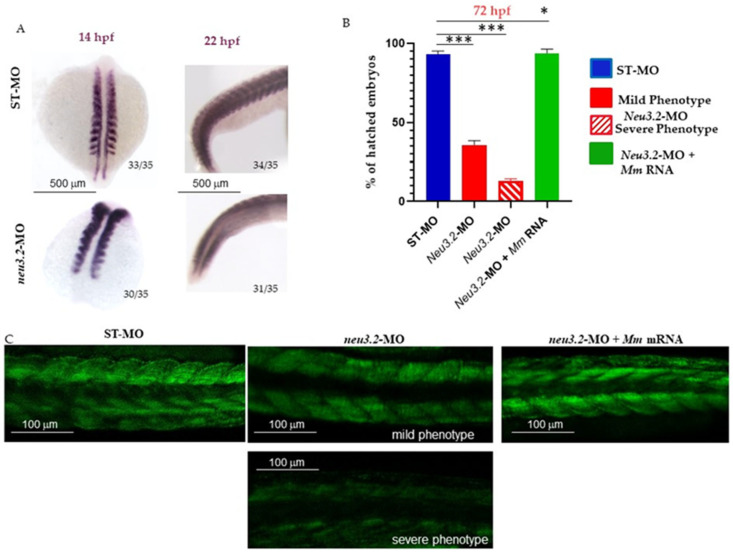
*neu3.2* knockdown causes defects in muscle formation. (**A**) Analysis of *myod1* expression in *neu3.2* morphants. Representative dorsal view images of the WISH analyses of *myod1* expression of 14 and 22 hpf embryos injected with the standard morpholino (STD-MO) and *neu3.2*-MO. Results are from one representative experiment with embryos (*n* = 35) out of two independent replicates. Ratios at the bottom-left part of each picture specify the number of embryos showing the same staining pattern, compared to the total number of embryos used for each experiment. (**B**) The graph shows the percentages of hatched embryos at 72 hpf after injection with *neu3.2* morpholino. Both categories were raised to 72 hpf and then analyzed for hatching. The *X*-axis shows the non-injected, injected with *neu3.2*-MO, with mild and severe phenotypes, and embryos rescued with murine Neu2 mRNA. The *Y*-axis reports the percentages of hatched embryos at 72 hpf. The results are expressed as the mean of 3 independent experiments, with 30 embryos for each experiment and each treatment. (* *p* and *** *p* < 0.001 vs. the control group—embryos injected with standard morpholino ST-MO). (**C**) Representative images with a light sheet microscope of Tg(*Bmp:EGFP*) embryos injected with the standard morpholino (STD-MO) and with *neu3.2*-MO (1.pmol/embryo) and rescued with murine Neu2 mRNA, observed at 48 hpf. A strong reduction in the fluorescence intensity is evident in *neu3.2* morphants, whereas a recovery is detectable upon rescue with murine Neu2 mRNA. The results are from one representative experiment with at least 25 embryos out of three independent replicates. Magnification 10X.

**Figure 6 ijms-24-13578-f006:**
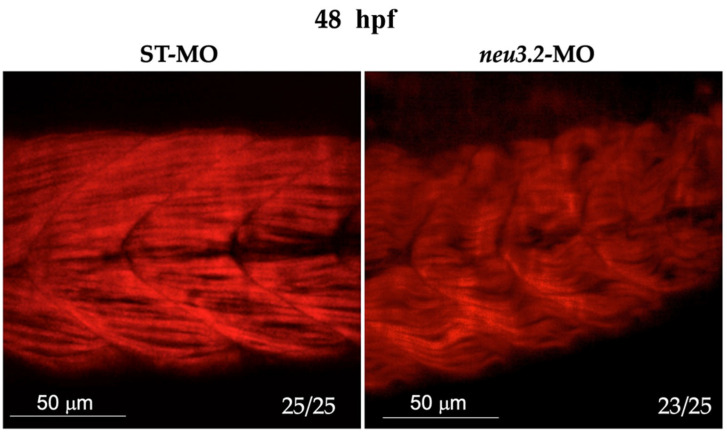
Knockdown of *neu3.2* interferes with myosin organization in slow myosin fibers and with myosepta. Light sheet imaging of the trunk longitudinal section after staining the actin fibers with phalloidin. Representative lateral views of the stained embryos showing a clear reduction in the total fluorescence intensity in *neu3.2* morphants. Images were acquired with the same parameters as described the Section 4. Ratios at the bottom-left part of each picture specify the number of embryos showing the same staining pattern, compared to the total number of embryos used for each experiment. The experiment was repeated twice with *n* = 25.

**Figure 7 ijms-24-13578-f007:**
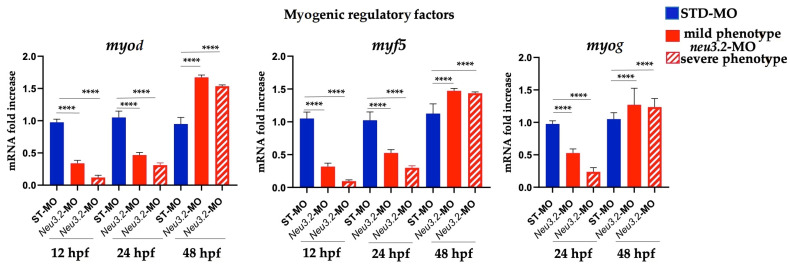
Expression level analysis of different markers involved in muscle development in neu3.2 morphants and embryos injected with the standard morpholino (STD-MO). Gene expression was normalized using *rpl13a* as a reference gene expressed as the mRNA fold increase. Data are representative of three replicates and are shown as the mean of standard deviation (**** *p* < 0.001 vs. injected embryos with standard morpholino-STD-MO).

**Figure 8 ijms-24-13578-f008:**
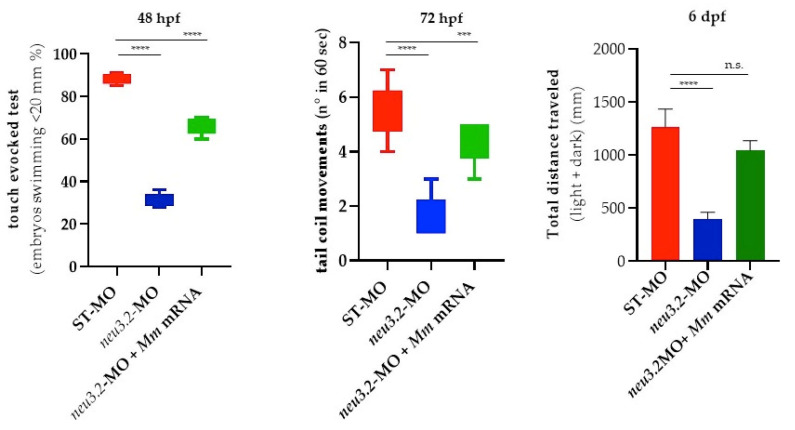
Locomotor analysis of neu3.2 morphants and murine Neu2-rescued embryos at different stages of development. Swimming performance of embryos at 48 hpf in the touch-evoked test. A significant progressive decrease in the percentage of *neu3.2* morphants when compared to embryos injected with standard morpholino (STD-MO) or rescued–injected embryos was detected (**** *p* < 0.0001, Student’s *t*-test test). The results are representative of three independent experiments with 40 embryos per group. At 72 hpf, a significant percentage of *neu3.2*-MO-injected embryos exhibited a decreased number of tail flip movements when compared to STD-MO or rescued *Neu2* mRNA-injected embryos (*** *p* < 0.001 vs. STD-MO; **** *p* < 0.0001 vs. STD-MO—Student’s *t*-test). At 6 dpf for a light–dark locomotion test, movements were reported as the mean ± SD of the total distance swam by the larvae (cm), calculated during both light and dark stimuli. The results are expressed as the mean ± SD of 3 independent experiments, with 12 surviving larvae for each experiment and for each treatment (**** *p* < 0.0001 vs. STD-MO; n.s.: not significative—Student’s *t*-test).

**Table 1 ijms-24-13578-t001:** Primers used for real-time PCR.

Primer Name	Forward	Reverse
*mif5*	5′-GAATAGCTACAACTTTGACG-3′	5′-GTAAACTGGTCTGTTGTTTG-3′
*myog*	5′-TCTGAAGAGGAGCACATTGA-3′	5′-AGCCCTGATCACTAGAGGA-3′
*myod1*	5′-TCAGACGAGAAGACGGAACA-3′	5-’CACGATGCTGGACAGACAAT-3′
*rpl13a*	5′-TCTGGAGGACTGTAAGAGGTATGC-3′	5′-AGACGCACAATCTTGAGAGCAG-3′

## Data Availability

Not applicable.

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
