# Peer review of "Downregulation of Zebrafish Cytosolic Sialidase Neu3.2 Affects Skeletal Muscle Development"

_ijms, 2023, doi:10.3390/ijms241713578_

Round 1
Reviewer 1 Report
Comments
Sialidase is an enzyme that releases sialic acid from the non-reducing ends of sugar chains and has various physiological functions. In this paper, the authors focus on the cytoplasmic sialidase Neu3.2 in fish and show that MO causes abnormalities in skeletal muscle. The results of this study are very significant because sialidase research in fish is limited. However, there are a few areas that need to be revised in order to be accepted.
1. line71: “Neu2 that appears only in birds and mammals”, but according to NCBI website, Neu2 genes are also found in turtles, alligators and amphibians. Authors should revise and discuss this.
2. In this paper, NI is used for comparison, not STD-MO. Why do authors dare to use NI when STD-MO is generally used because of the physical effects of injection?
3. Line149 also has "Figure S3, most of .... .while 4% of the morphants showed the mild one", but Figure S3 only lists the mortality rate, which does not fit the description.
4. Line 509 Please show the primer sequence of rpl13a in Table 1.
5. Scale bar is required for all photos.
Author Response
Response to reviewer 1
1) line71: “Neu2 that appears only in birds and mammals”, but according to NCBI website, Neu2 genes are also found in turtles, alligators and amphibians. Authors should revise and discuss this.
We apologize for the oversight and we thank reviewer 1 for pointing it out. The text of the revised version of the manuscript has been changed as requested.
2) In this paper, NI is used for comparison, not STD-MO. Why do authors dare to use NI when STD-MO is generally used because of the physical effects of injection?
The authors thank reviewer 1 for raising this important point. The revised version of the manuscript now contains only results obtained from SDT-MO injected fishes as control.
3) Line149 also has "Figure S3, most of .... .while 4% of the morphants showed the mild one", but Figure S3 only lists the mortality rate, which does not fit the description.
The authors apologize for the oversight and thank reviewer 1 for raising this point. The paragraph 2.3 has been strongly modified and now the results reported in figures S3, S4 and 2 are sequentially described.
4) Line 509 Please show the primer sequence of rpl13a in Table 1.
The forward and reverse sequences of the oligonucleotides specific for rpl13a are now reported in Table 1.
5) Scale bar is required for all photos.
Again, the authors apologize for the oversight and thank reviewer 1 for highlighting it. All the photos of the revised version of the manuscript contain the scale bar.
Reviewer 2 Report
The manuscript 'Downregulation of zebrafish cytosolic sialidase neu3.2 affects 2 skeletal muscle development' is well written and the study was performed thoroughly.
Please find the following comments which could be useful.
1. The introduction could have been better explained why this study was conducted and what is the knowledge gap in this area?
2. In figure 1, why 9 month old adult was selected? any rationale?
3. Line 129-130 - The statement is misleading as high sialidase activity was found in both ovary and muscle.
4. Line 153 - Typo error
5. Line 196 - Hearth?
6. Figure 5C - Can you replace better NI image?
Well written article with a few typos.
Author Response
Response to reviewer 2
1) The introduction could have been better explained why this study was conducted and what is the knowledge gap in this area?
The authors thank reviewer 2 for this important suggestion. In the revised version of the manuscript, introduction have been modified accordingly by introducing new sentences and reference.
- In figure 1, why 9 month old adult was selected? any rationale?
The choice is linked to the high sialidase activity detectable starting from 6 months (preliminary experiments) and at 9 months the animal is in the maximum phase of its adult life.
- Line 129-130 - The statement is misleading as high sialidase activity was found in both ovary and muscle.
The authors thank reviewer 2 for highlighting this point. It was an oversight and the sentence in the revised version of the manuscript has been modified according to the results of Figure 1B.
- Line 153 - Typo error
Again, the authors thank reviewer 2 for the careful revision and apologize for the typo. The sentence has been modified accordingly.
Line 196 - Hearth?
We apologize for the typo. The revised version has been modified and the term is now heart.
Figure 5C - Can you replace better NI image?
Figure 5C has been modified with STD-MO image. We hope this one will be of better quality.
